# Evaluation of the Hydraulic, Physical, and Mechanical Properties of Pervious Concrete Using Iron Tailings as Coarse Aggregates

**Yan Tan** [ID], **Yuntao Zhu** and **Henglin Xiao** *

Department of Structural Engineering, School of Civil Architecture and Environment, Hubei University of Technology, Wuhan 430068, China; tanyan@hbut.edu.cn (Y.T.); zhuyuntao0725@163.com (Y.Z.)
* Correspondence: xiaohenglin@hbut.edu.cn

**Abstract:** Aggregates are a significant component of pervious concrete. Pervious concrete using iron tailings as coarse aggregates was prepared to study the feasibility of this approach. A mix design procedure was also used to design pervious concrete based on different target porosities. The effective porosity, measured porosity, dry density, compressive strength, and permeability coefficient of pervious concrete were studied. The results show that the mix design procedure based on a target porosity is relatively reasonable for designing iron tailing-based pervious concrete. The 28 d compressive strength of the pervious concrete decreased from 42 to 11 MPa as the effective porosity increased from 5.2% to 27.2%. The effective porosity of the equilibrium point of the compressive strength and permeability coefficient of pervious concrete was approximately 16%, where the compressive strength was 21.5 MPa, and the water permeability was 3.2 mm/s. The permeability coefficient of pervious concrete can be predicted as an exponential function of the effective porosity, and the compressive strength of pervious concrete can be predicted as a logarithmic function of the effective porosity.

**Keywords:** iron tailings; pervious concrete; porosity; permeability coefficient

## 1. Introduction

Pervious concrete is one of the most typical components in the construction of sponge cities. The purpose of sponge cities is mainly to alleviate the heat island effect and to improve the hydrological environment of the city [1,2]. Pervious concrete is a kind of special porous concrete that is not prepared by physical or chemical foaming but by using gap-graded aggregates, usually without fine aggregates. Pervious concrete has a large number of tortuous and interconnected pores. Rainwater can be drained off into the groundwater through these pores [3,4]. Although the spatial structure of a pore, such as pore size, tortuosity, connectivity, and pore distribution, has an influence on the permeability coefficient, strength and durability of the pervious concrete to a certain extent, the porosity is still the controlling factor of these characteristics [5,6]. Therefore, the current mix design methods of pervious concrete are based on the porosity [7,8]. However, the design method of Nguyen's permeable concrete is too complex to adapt to the actual engineering environment [7]. The design method of CJJ/T-135 is too simple [8], and the theoretical results are quite different from the actual results. Meanwhile, it is expected that a sufficient supply of aggregates has a significant impact on the development of pervious concrete. Therefore, China's restrictions on the exploitation of natural ores have created favorable conditions for the utilization of iron tailings [9]. Iron tailings are a kind of mining waste produced from the concentration of iron ore in the beneficiation process [10]. At present, there are approximately 5 billion tons of iron tailings in China, and the annual growth rate

is approximately 600 million tons [11]. However, the utilization rate of iron tailings is only 7% [12]. Iron tailings are mainly used for recovery [13], fire blocks [14], ceramsite [15], concrete aggregate [16], mineral admixture in the process and concrete industry [17], and additions in the ceramic industry [18] and other fields. Work by Ma [19] studied the heavy metal leaching of iron tailings, which shows that the concentration of heavy metal leaching of iron tailings is within the range of Chinese standards of GB/T 5085.3 [20]. Table 1 summarizes the variables and properties of pervious concrete using different aggregates [21–28]. We can draw two conclusions from Table 1: (1) The substitution of solid waste for the natural aggregate of pervious concrete has attracted much attention. (2) The variables of the studies mainly focus on the replacement amount and aggregate size as well as the binder type and curing conditions. Although investigations on pervious concrete have been reported extensively, there are few reports on the feasibility of iron tailings as coarse aggregates in pervious concrete. In this paper, iron tailings-based pervious concrete with different target porosities was prepared by a mix design procedure to study the feasibility of iron tailings as coarse aggregates. The effective porosity, measured porosity, dry density, compressive strength and permeability coefficient of the pervious concrete were measured and analyzed to obtain the key technologies for preparing iron tailings-based pervious concrete.

**Table 1.** Summary of the variables and properties of pervious concrete.

| Cementitious Materials | Aggregate Types | Variables | Porosity (%) | Permeability (mm/s) | Compressive Strength (MPa) | Reference |
|---|---|---|---|---|---|---|
| 43-grade Ordinary Portland 44-cement | Over burnt brick | aggregate size | 8~31 | 7~19 | 7~30 | [21] [a] |
| ordinary Portland cement (OPC) | Natural aggregates and recycled concrete aggregates (RCA) | RCA replacement | 18~22 | 8~20 | 3~17 | [22] [b] |
| Ordinary Portland Cement | basalt aggregates | aggregate size | 20.50~21.16 | – | 19.86~32.00 | [23] |
| alkali activated materials based on slag and/or metakaolin | natural gravel | aggregate size and binder type | 17.7~34.5 | 3.2~16.2 | 11.3~36.2 | [24] |
| type I Portland cement | dolomite aggregate and copper slag | Copper slag replacement | 20.44~22.77 | 3.05~3.53 | 17.93~23.45 | [25] |
| magnesium phosphate cement | waste steel slag | aggregate size | 25~26 | 6.8~7.1 | 34~38 | [26] [c] |
| type I cement, type II cement, sulphoaluminate cement and calcium aluminate cement | electric arc furnace slag | cement type and curing condition | 8.45~21.31 | 4.52~7.62 | 10.02~24.72 | [27] [d] |
| 53-grade Ordinary Portland Cement, class-F type fly-ash | limestone aggregates | fly ash content | 18~35 | 6.1~9.8 | 5.0~10.8 | [28] [e] |

[a]: The values detected from observations of the figures where w/c = 0.3 and fine aggregate = 10%. [b]: The values obtained from P20-S0. [c]: The values detected from observations of the figures for the tamping molding. [d]: The values obtained for Type I cement, Type II cement, and calcium aluminate cement cured in saturated lime water. [e]: The values detected from observations of the figures.

## 2. Materials and Methods

### 2.1. Materials

The cement conforming to GB/T 175 [29] was P.O. 42.5 grade Portland cement (OPC) produced by Tangshan Jidong Cement Co., Ltd. (Hebei, China). Iron Tailings were used as the pervious concrete coarse aggregates, with selected particle size of 4.75–9.5 mm as the test material (Liaoning, China). The basic properties of the coarse aggregate of iron tailings are shown in Table 2. The water reducing agent was the powder polycarboxylate high performance water reducing agent supplied by Dongfang Yuhong Co., Ltd. (Beijing China). The water reduction rate was ≥28%. Silica fume with a specific density of 2.2 g/cm$^3$ was collected from a silicon iron smelter.

**Table 2.** Basic properties of the coarse aggregate of iron tailings.

| Category | Particle Size (mm) | Specific Gravity (kg·m$^{-3}$) | Bulk Density (kg·m$^{-3}$) | Void Ratio (%) |
|---|---|---|---|---|
| Iron tailing | 4.75–9.5 | 2750 | 1500 | 45 |

### 2.2. Mix Design

Pervious concrete is similar to ordinary concrete in composition, which is composed of cementitious material, aggregate, water and an admixture. The difference is the amount of fine aggregate in pervious concrete. The pervious concrete in this paper does not use fine aggregates. The mix design of pervious concrete is a decisive factor for determining the performance of pervious concrete. Usually, a target porosity is set first, and then the mass of the corresponding components are calculated by the following equations.

$$W_A = \alpha \cdot \rho_A \tag{1}$$

$$V_{CM} = 1 - \alpha \cdot (1 - \varphi_A) - \varphi_{TP} \tag{2}$$

$$W_C = \frac{\rho_C \rho_W V_{CM}}{\rho_C R_{w/c} + \rho_W} \tag{3}$$

$$W_w = W_c \cdot R_{w/c} \tag{4}$$

$$W_{wr} = \delta \cdot W_C \tag{5}$$

$$W_{MA} = \left( V_{CM} - \frac{W_W}{\rho_W} - \frac{W_c(1 - \beta)}{\rho_C} \right) \cdot \rho_{MA} \tag{6}$$

where: $W_A$ is the mass of the aggregate of pervious concrete measured in unit volume, kg/m$^3$. $\alpha$ is the correction coefficient of the aggregate usage, taking 0.98. $\rho_A$ is the bulk density of the aggregates, kg/m$^3$. $V_{CM}$ is the paste volume of the cementitious materials, m$^3$/m$^3$. $\varphi_A$ is the packed porosity of the aggregates, %. $\varphi_{TP}$ is the target porosity of the pervious concrete, %. $W_C$ is the mass of the cement in unit volume, kg/m$^3$. $R_{w/c}$ is the ratio of the water to the cementitious materials. $\rho_C$ is the true density of the cement, taking 3150 kg/m$^3$. $W_w$ is the mass of the water in unit volume, kg/m$^3$. $W_{wr}$ is the mass of the water reducer in unit volume, kg/m$^3$. $\delta$ is the ratio of the water reducer to the cementitious materials. $W_{MA}$ is the mass of the mineral admixture in unit volume, kg/m$^3$. $\rho_W$ is the true density of water at 20 °C, taking 1000 kg/m$^3$. $\rho_{MA}$ is the true density of the mineral admixture, kg/m$^3$.

In this paper, different target porosities (8%~30%) were selected to prepare pervious concrete using iron tailings as the coarse aggregates. At the same time, the water-cement ratio, silica fume-cement ratio, and water reducer-cement ratio were 0.23%, 6%, and 0.2%, respectively. For clarity, the mix proportions of different pervious concrete mixture are shown in Table 3.

**Table 3.** Mix ratio design of pervious concrete under different target porosities.

| Target Porosity % | Cement kg·m⁻³ | Aggregate kg·m⁻³ | Silica Fume kg·m⁻³ | Superplasticizer kg·m⁻³ | Water kg·m⁻³ |
|---|---|---|---|---|---|
| 8.0 | 666.8 | 1470 | 29.2 | 1.39 | 160 |
| 10.0 | 631.8 | 1470 | 27.6 | 1.32 | 152 |
| 12.5 | 588.0 | 1470 | 25.7 | 1.23 | 141 |
| 15.0 | 544.3 | 1470 | 23.8 | 1.14 | 131 |
| 17.5 | 500.5 | 1470 | 21.9 | 1.04 | 120 |
| 20.0 | 456.8 | 1470 | 20.0 | 0.95 | 110 |
| 22.5 | 413.0 | 1470 | 18.1 | 0.86 | 99 |
| 25.0 | 369.3 | 1470 | 16.2 | 0.77 | 89 |
| 27.5 | 325.5 | 1470 | 14.2 | 0.68 | 78 |
| 30.0 | 281.8 | 1470 | 12.3 | 0.59 | 68 |

*2.3. Sample Preparation*

The processes of sample preparation for pervious concrete are illustrated in Figure 1. The cement, silica fume and water reducer were first poured into a blender with a 140 r/min rotation speed, and the mixture was stirred for 2 min. Next, iron tailings were added, and the mixture was stirred for another 2 min. Then, water was added, and the mixture was stirred for 5 min to obtain a consistent mixture. The obtained mixture was cast in three $100 \times 100 \times 100$ mm cubes for the strength test and in three $\Phi100 \times 100$ mm cylinders for the permeability and porosity test. A compactive effort of five drops of Proctor hammer per layer for two layers in the molds was applied. The bottom diameter, the mass and the fall height of the Proctor hammer are 51 mm, 2.5 kg, and 305 mm, respectively, and the average compaction energy is approximately 200 kN·m/m³. These samples were sealed with a polythene sheet to avoid water evaporation and cured under ambient conditions for 1 d, demolded, and further cured in a standard curing tank at a temperature of $20 \pm 5$ °C and a humidity of more than 90% for another 27 d for relevant tests.

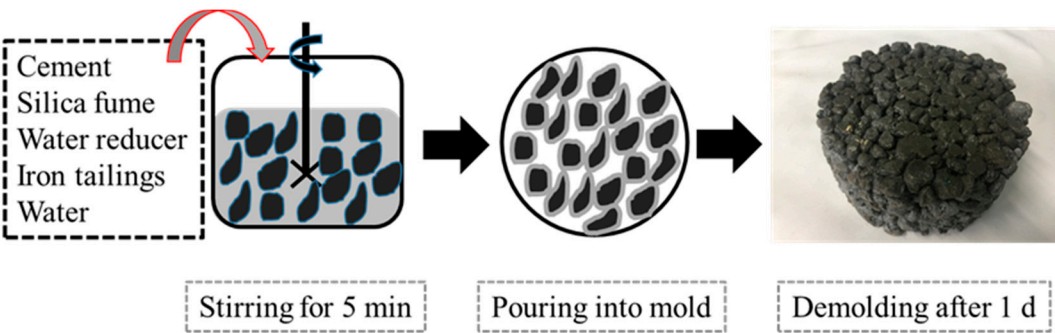

**Figure 1.** Preparation processes for pervious concrete.

*2.4. Methodology*

2.4.1. Effective Porosity and Measured Porosity

The effective porosity and measured porosity were determined on cylinders that were dried in an oven at 50 °C until reaching a constant weight and then cooled to room temperature. The effective porosity, also called open porosity, was calculated according to Equation (7). This method has been described in the literature [25,26]:

$$P_e = \left[ 1 - \frac{W_0 - W_1}{V_0 \cdot \rho_W} \right] \times 100\% \tag{7}$$

where:$P_e$ is the effective porosity of the pervious concrete. $W_0$ is the dry mass of the pervious concrete, kg. $W_1$ is the mass of the pervious concrete saturated in water, kg. $V_o$ is the bulk volume of the pervious concrete, $m^3$. $\rho_W$ is the true density of water at 20 °C, taking 1000 kg/$m^3$. The measured porosity was determined by image analysis based on Image-Pro Plus software. This procedure, as shown in Figure 2, has been reported in many studies [25,30]. Nine images of the three fracture sections (top, middle, and bottom sections) of each specimen were captured by the digital camera. Then, the black/white binarization and the measured porosity of each image were determined by the software.

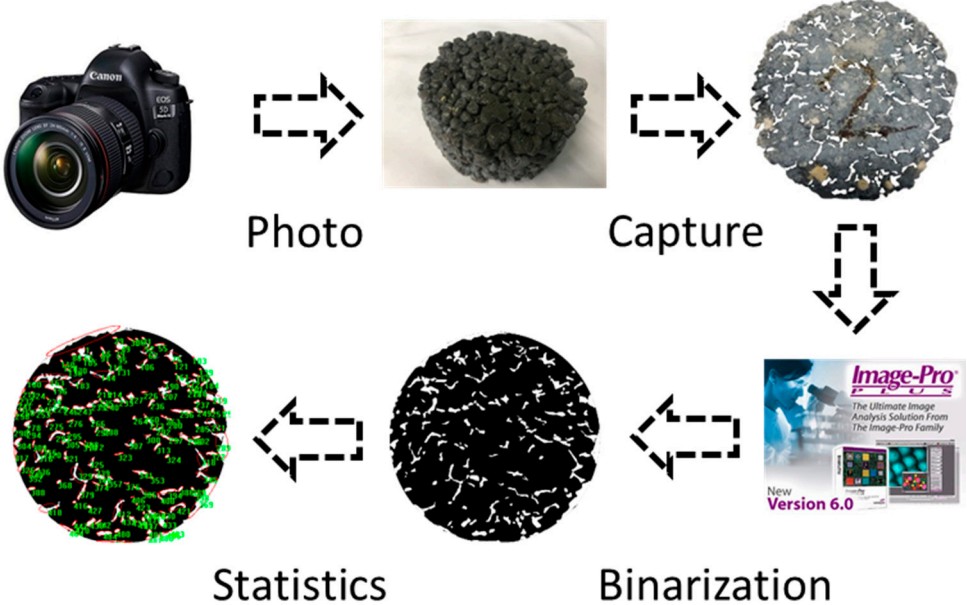

**Figure 2.** Schematic of the porosity determination processes of specimens using Image-Pro Plus.

### 2.4.2. Bulk Density and Compressive Strength

The bulk density was determined on cubes that were dried in an oven at 50 °C until reaching a constant weight and then cooled to room temperature. The bulk density of the pervious concrete was calculated by dividing the dry mass by the bulk volume of the pervious concrete. The compressive strength was tested at 28 d after casting according to the Chinese standard GB/T 50081, as shown in Figure 3. The values of the bulk density and compressive strength were the average of three valid measurements.

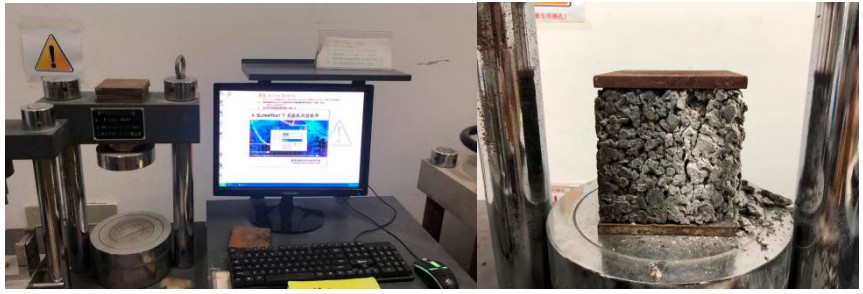

**Figure 3.** Compressive strength test.

### 2.4.3. Permeability Coefficient

The permeability coefficient is one of the most significant indicators to evaluate pervious concrete. A Constant falling-head apparatus, as shown in Figure 4, was used to determine the permeability coefficient of the pervious concrete. Each specimen was sealed with petroleum jelly and put into a latex

membrane to prevent leakage along the sides during testing. A specimen with a cross-sectional area of *A* and a height of *L* was put into transparent plastic cylinder. Then, the water valve was opened to let the water flow through the specimen. After the head difference Δh and the flow *Q* was stable, the water volume V flowing through the sample within a certain period of time *t* was recorded. The measurement was repeated three times for each specimen to obtain a mean value. The permeability coefficient k is calculated according to Equation (8). A similar approach was adopted by the literature [26,27,31].

$$k = \frac{Q \times L}{A \times \nabla h \times t} \tag{8}$$

where: k is the water permeability coefficient (mm/s); *Q* is the discharged amount of water at time *t* (ml); *L* is the thickness of the sample (mm); *A* is the upper surface area of the sample (mm$^2$); $\nabla h$ is the water head difference, taking 150 mm; and t is the passing time (s), taking 300 s.

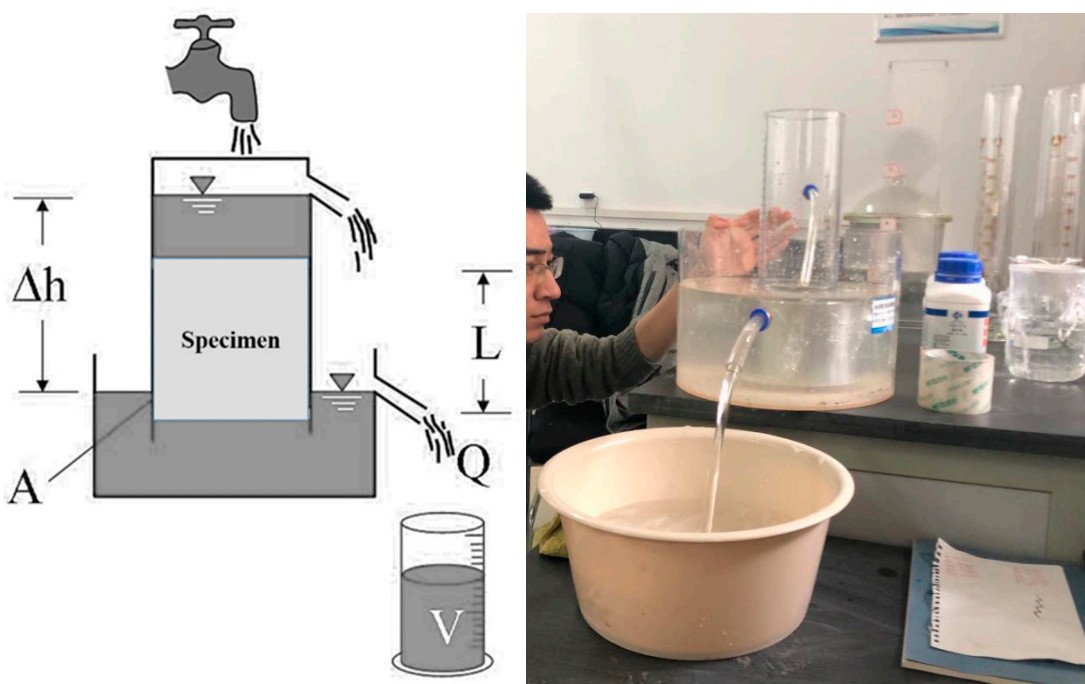

**Figure 4.** Constant falling-head permeability setup.

## 3. Results and Discussions

### 3.1. Measured Porosity and Effective Porosity of Pervious Concrete

The values of properties of pervious concrete are the average values of the three specimens. The error bar is the standard deviation of the values of the three specimens. The shorter the error bar is, the smaller the experimental error is, and the more accurate the results are. The effective porosity and measured porosity of the pervious concrete under different target porosities are represented in Figure 5. As shown in Figure 5, the measured porosity of the pervious concrete ranged from 7.05% to 29.32%, which is slightly more than the effective porosity ranging from 5.12% to 27.14% when the target porosity of pervious concrete was designed from 8% to 30%. These differences indicated that part of the designed pores were compacted with hydration products, called 'compacted pores', or surrounded and semisurrounded by hydration products, called closed or semiclosed pores. Figure 6 presents the schematic of the component and structure of the pervious concrete. Pervious concrete is composed of hydration products, iron tailings, compacted pores, closed pores, and open pores. The number of open pores is the controlling factor of the water permeability of pervious concrete. The water on the upper surface of pervious concrete can only flow through the open pores to the lower surface of pervious

concrete but cannot flow through the closed pores or semiclosed pores, which can be measured by Image-Pro Plus. Apart from the test error, this is the main reason for the difference between the effective porosity and measured porosity of the pervious concrete. Although there is a certain gap between the effective porosity and the target porosity, within a certain correction range, it shows that the design formula of the pervious concrete in this paper is reasonable and can, to a certain extent, be used as a reference. Although there is a gap between the measured porosity and target porosity, ranging from 1.2% to 2.4%, it only accounts for 5% to 10% of the target porosity and these values were within the design error margin. Therefore, the design method of pervious concrete used here was, to a certain extent, reasonable and was feasible for pervious concrete using iron tailings as coarse aggregates.

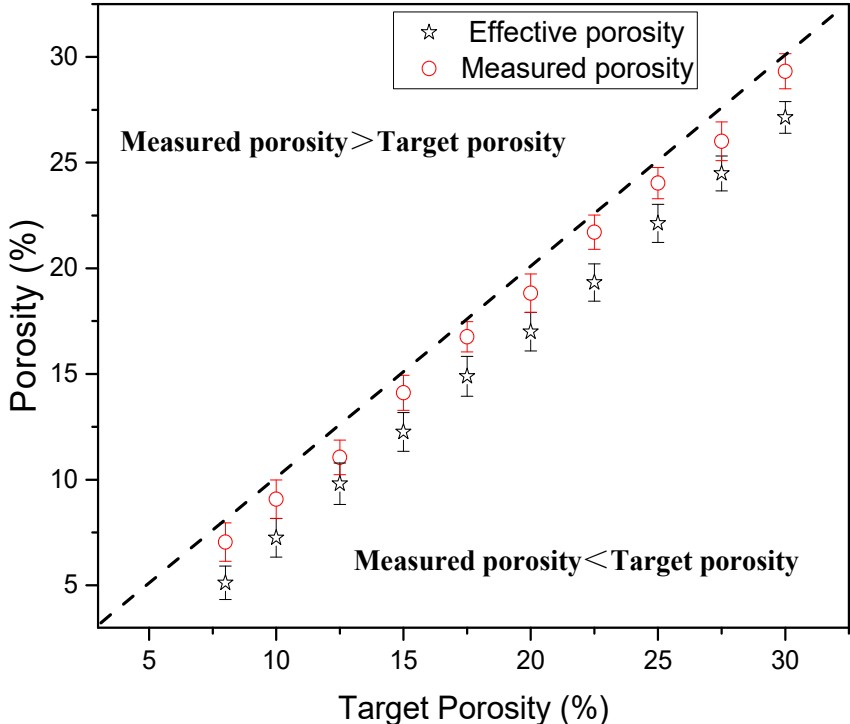

**Figure 5.** Effective porosity and measured porosity of the pervious concrete with different target porosities.

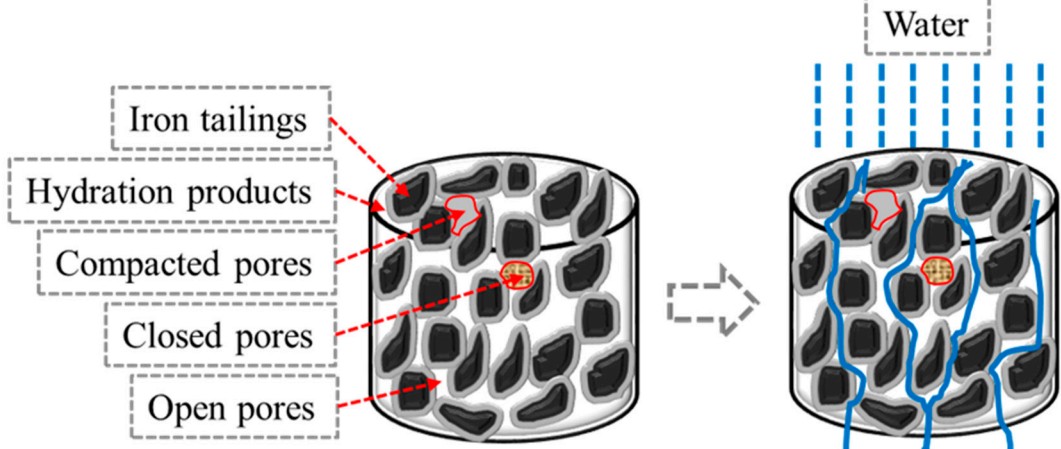

**Figure 6.** Schematic of the component and structure of pervious concrete.

### 3.2. Dry Density of Pervious Concrete

Figure 7 displays the dry density of the pervious concrete and its relationship versus the effective porosity and measured porosity. As shown in Figure 7, the dry density of the pervious concrete decreased from 2240 to 1798 kg/m³ with increasing target porosity from 8% to 30%, which is typical for pervious concrete (1600~2200 kg/m³) [22]. The relationships between the dry density and the effective porosity and measured porosity are represented in Equations (9) and (10), respectively. In practical engineering applications, the relationship between the dry density and effective porosity is recommended as the optimal equation, not only due to its high correlation coefficient, but also due to the relatively ease for determining the value of the effective porosity compared to measured porosity.

$$\rho_d = 15.86 \cdot P_e + 2242 \quad R_2 = 0.915 \tag{9}$$

$$\rho_d = 15.58 \cdot P_m + 2266 \quad R_2 = 0.905 \tag{10}$$

where: $\rho_e$ is the dry density of pervious concrete, kg/m³; $P_e$ is the effective porosity, %; and $P_m$ is the measured porosity, %.

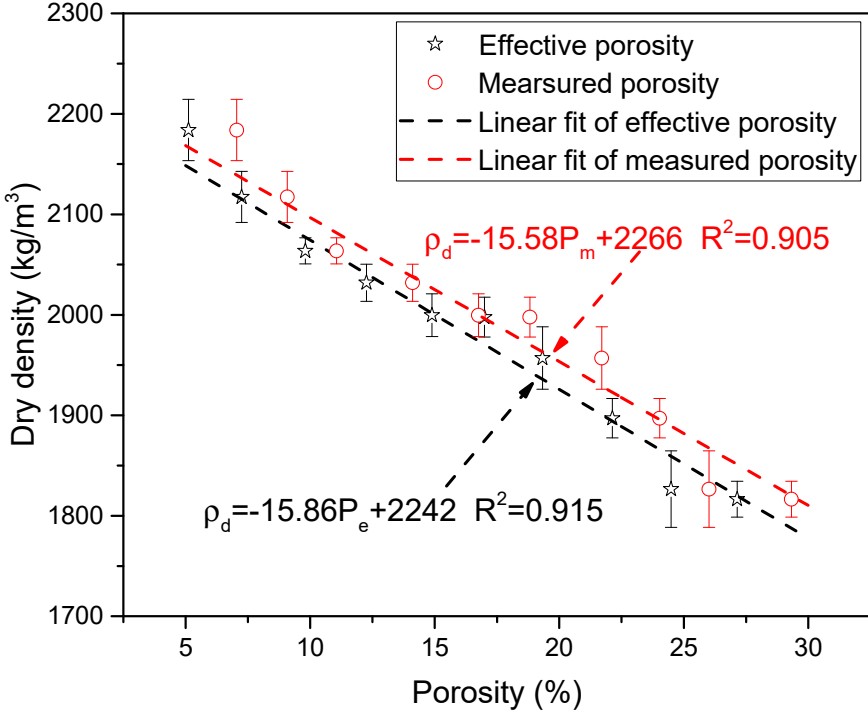

**Figure 7.** Dry Density of the pervious concrete and its relationship versus the effective porosity and measured porosity.

### 3.3. Compressive Strength of Pervious Concrete

The 28 d compressive strength of the pervious concrete and its relationships versus the effective porosity are shown in Figure 7. It is expected that the 28 d compressive strength of the pervious concrete decreased from 42 MPa to 11 MPa with the effective porosity increasing from 5.2% to 27.2%. It has been reported that the porosity has a decisive effect on the strength of concrete, at the same time, the pore structure (such as pore size, pore distribution, pore shape and orientation, etc.) also affects the compressive strength of the concrete [32,33]. Work by M. O'Farrell [32] shows that there is a critical relationship between compressive strength and pore threshold radius of mortar. As threshold radius approaches gel pore size, compressive strength increases rapidly, indicating that pore size, rather than pore volume, is a limiting factor of compressive strength development. Rakesh's study [33] shows that

existing linear functions relating the strength with pore size characteristics of cement-based material are inadequate in the context of concrete. Therefore, the compressive strength and effective porosity of concrete do not exhibit a simple linear function. The relationship between the compressive strength and effective porosity is fitted with different functions (see Figure 8). Among these, the correlation of the logarithmic function was best. Therefore, the logarithmic function, displayed as $f_c = 72.9 - 18.4 \times \ln(P)$, was selected to analyze and predict the compressive strength of pervious concrete under different effective porosities.

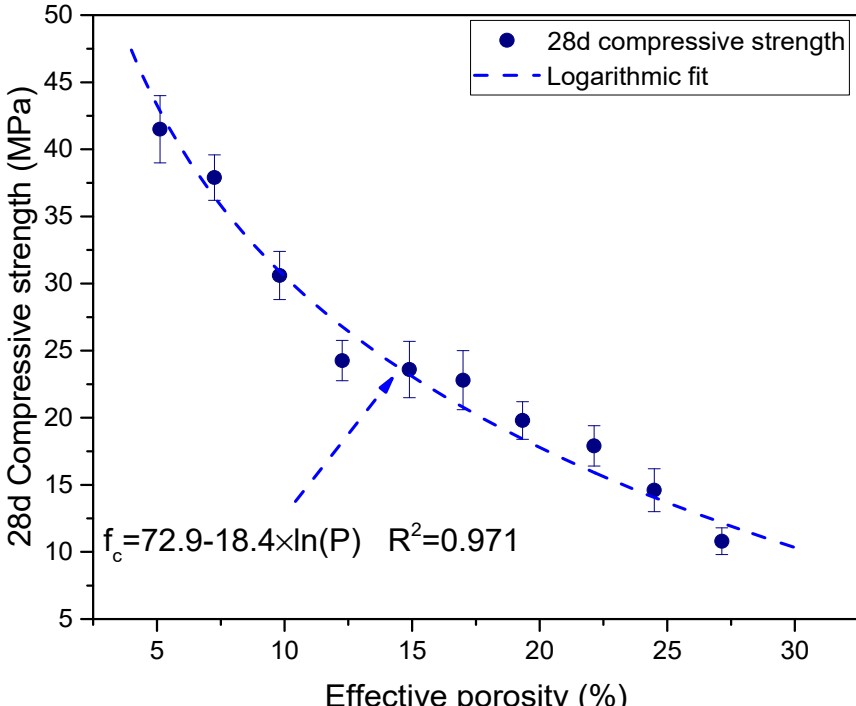

**Figure 8.** The 28 d compressive strength of the pervious concrete and its relationships versus the effective porosity.

### 3.4. Permeability Coefficient of Pervious Concrete

Figure 9 illustrates the permeability coefficient of the pervious concrete and its relationships versus the effective porosity. As shown in Figure 9, the permeability coefficient of pervious concrete increased from 0.67 to 8.2 mm/s as the effective porosity increased from 5.2% to 27.2%. According to the literature [4,34–36], the relationship between the permeability coefficient and effective porosity of pervious concrete can be divided into four functions: (1) the classical Kozeny-Carman function; (2) the power function; (3) the exponential function; and (4) monotonic linear function. Comparing the fitting results of the different functions, the exponential function had the highest relevance of fit. This result was also consistent with previous studies [37]. Therefore, it is reasonable and relatively accurate to use the exponential function, displayed as $k = 0.57 \times e^{0.98P}$, to analyze the relationship between the permeability coefficient of pervious concrete using iron tailings as coarse aggregate. The boundaries of applicability for the compressive strength and permeability coefficient functions obtained in Figures 8 and 9 can be summarized as following: (1) P.O. 42.5 grade Portland cement is used as cementitious materials, (2) the water to cement mass ratio is about 0.23 (3) the compaction method is a compact effort of five drops of Proctor hammer per layer for two layers in the molds, and (4) the aggregate size is 4.75–9.5 mm.

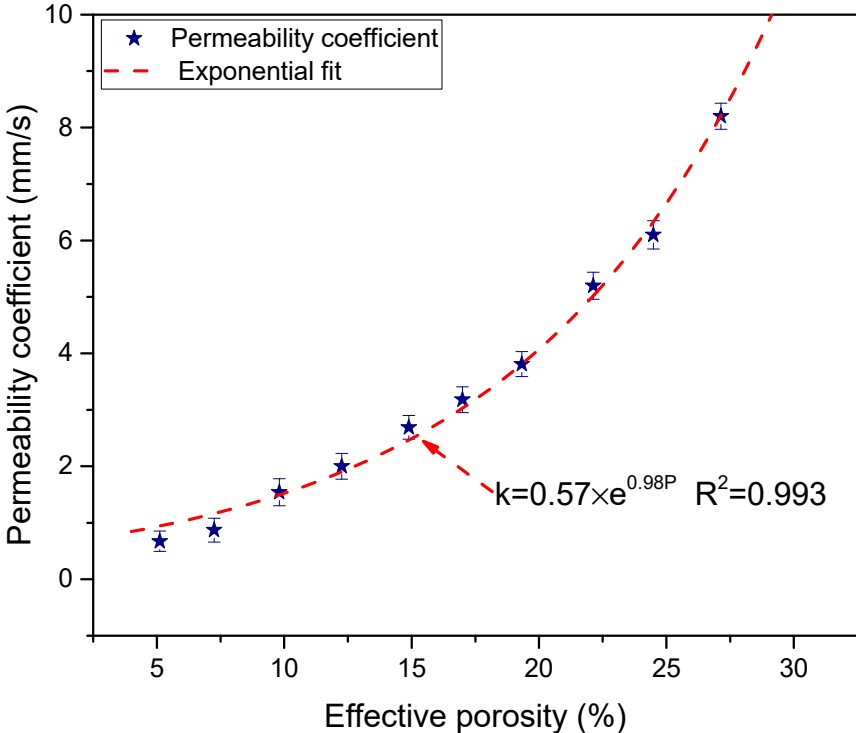

**Figure 9.** Permeability Coefficient of the pervious concrete and its relationships versus the effective porosity.

### 3.5. Equilibrium Point of the Permeability and Strength of Pervious Concrete

Figure 10 shows the equilibrium point of the compressive strength and permeability coefficient variation with the effective porosity of the pervious concrete. The permeability coefficient at 1 mm/s is the minimum value for the pervious concrete [38]. The pervious concrete with a permeability coefficient with above 1 mm/s can be used as a functional zone. Otherwise, the pervious concrete is called nonfunctional pervious concrete. Meanwhile, according to the compressive strength, functional pervious concrete can be divided into two types of pervious concrete [38]. When the compressive strength of pervious concrete is above 20 MPa, the pervious concrete is usually used in pavement zone. When the compressive strength of pervious concrete is below 20 MPa the pervious concrete is usually used at nonpavement zone, such as garden and greenbelt zones. The effective porosity of the equilibrium point of the compressive strength and permeability coefficient of the pervious concrete was approximately 16%, where the compressive strength was 21.5 MPa and the water permeability was 3.2 mm/s. It is worth noting that there is a difference between the effective porosity and target porosity. Therefore, when the mix design procedure here is used for actuarial engineering, it would be appropriate to increase the value of the target porosity. The optimum porosity ranged from 7.5% to 16.5%, which provides a theoretical basis for the mix design and performance for the preparation of pervious concrete.

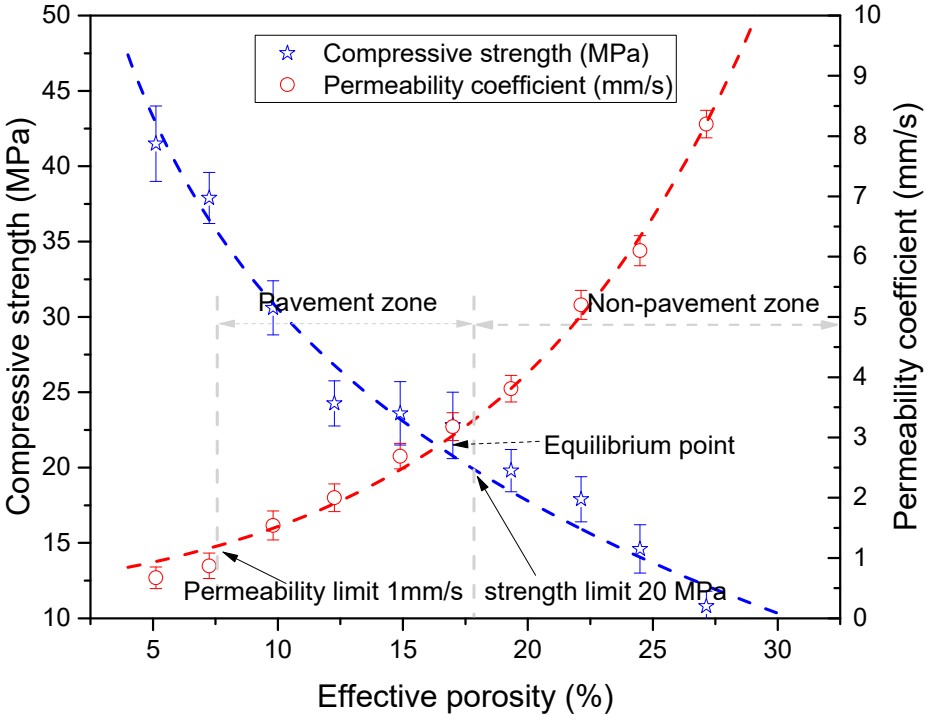

**Figure 10.** Equilibrium point of the compressive strength and permeability coefficient variation with effective porosity of the pervious concrete.

*3.6. Comparisons of the Permeability Coefficient and Compressive Strength of Pervious with Different Types of Aggregates*

Figure 11 shows the comparisons of the permeability coefficient (A) and compressive strength (B) of pervious concrete with different types of aggregates [21–28]. Although different experimental conditions such as porosity type (fresh state, total, or open), test method and samples size may restrict the comparison of the properties of pervious with different types of aggregates, Some useful information can still be found by summarizing and analyzing these pervious concretes. As shown in Figure 11, compared with other types of aggregates, the values of both the permeability coefficient and compressive strength of iron tailings-based pervious concrete described here were located in the moderate position. For example, when the porosity is at the typical value of 15%~30%, the permeability coefficient and compressive strength of the iron tailings-based pervious concrete are 2~10 mm/s and 10~25 MPa, respectively, which are similar to the values of the pervious concretes using natural gravel, copper slag, electric arc furnace slag, and limestone as aggregates. In contrast, the compressive strengths of burnt brick-based pervious concrete and recycled concrete-based pervious concrete are sacrificed for higher permeability coefficients. For pervious concrete using the same ordinary Portland cement, such as burnt brick-based pervious concrete, recycled concrete-based pervious concrete and iron tailings-based pervious concrete (see Table 1), although their compressive strength are similar under the same porosity, but the permeability coefficient of burnt brick-based pervious concrete and recycled concrete-based pervious concrete is one order of magnitude higher than that of iron tailings-based pervious concrete, which may be mainly due to the good pore structure of burnt brick-based pervious concrete and recycled concrete-based pervious concrete, with fewer semi-connected pores.

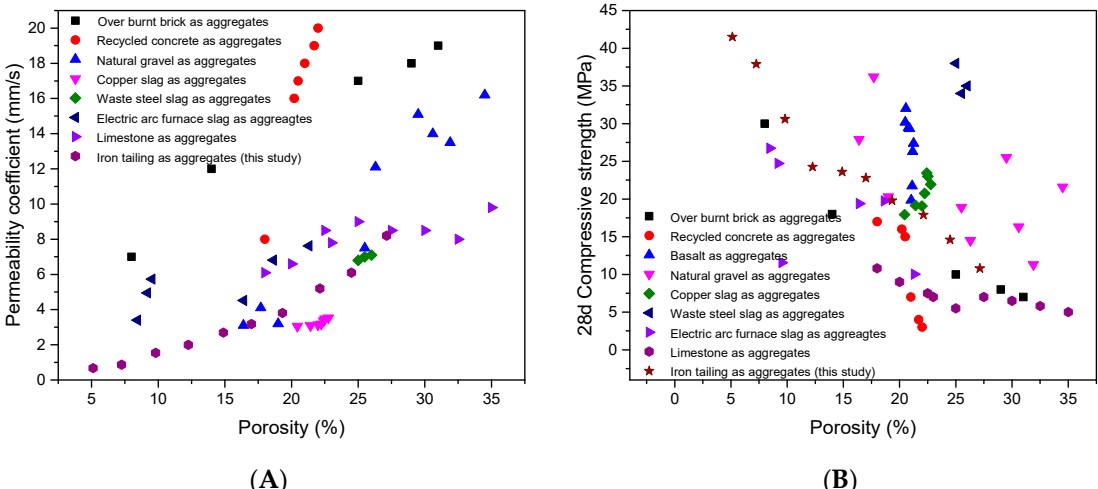

**Figure 11.** Comparisons of the permeability coefficient (**A**) and compressive strength (**B**) of the pervious concrete with different types of aggregates.

## 4. Conclusions

(1) According to a mix design procedure, when the target porosity of pervious concrete was designed from 8% to 30%, the measured porosity of the pervious concrete ranged from 7.05% to 29.32%, while the effective porosity ranged from 5.12% to 27.14%. It is relatively accurate to utilize the mix design procedure for pervious concrete using iron tailing as coarse aggregates.

(2) The dry density of the pervious concrete, ranging from 2240 to 1798 kg/m$^3$, was linearly distributed with both the measured porosity and effective porosity. The 28 d compressive strength of the pervious concrete decreased from 42 to 11 MPa as the effective porosity increased from 5.2% to 27.2%. The relationship between the dry density and effective porosity can be expressed as $\rho_d = 15.86 \times P_e + 2242$, while the relationship between the compressive strength and effective porosity can be expressed as $f_c = 72.9 - 18.4 \times \ln(P)$.

(3) The limitation and conditions for the compressive strength and permeability coefficient functions can be summarized as following: (1) P.O. 42.5 grade Portland cement is used as cementitious materials, (2) the water to cement mass ratio is about 0.23, (3) the compaction method is a compact effort of five drops of Proctor hammer per layer for two layers in the molds, and (4) the aggregate size is 4.75–9.5 mm.

(4) The permeability coefficient of pervious concrete, ranging from 0.67 to 8.2 mm/s can be predicted by as a function of the effective porosity, as displayed $k = 0.57 \times e^{0.98P}$. The effective porosity of the equilibrium point of the compressive strength and permeability coefficient of pervious concrete was approximately 16%, where the compressive strength was 21.5 MPa and the water permeability was 3.2 mm/s. The products prepared here can be used in parks, gardens, open parking lots, sidewalks, and some riverbanks.

**Author Contributions:** Y.T. is responsible for the idea providing and the paper revision. Y.Z. is responsible for the experimental process and the manuscript writing. H.X. is responsible for the experiment directing. All authors have read and agreed to the published version of the manuscript.

**Funding:** This research was funded by the Major Technological Innovation Program of Hubei Province (Grants no. 2018AAA028), National Natural Science Foundation of China (NSFC) [Grants no.51408203], Green Industry Project of Hubei University of Technology (Grants no.BSQD12060).The authors would like to express their appreciation to these financial assistances.

**Acknowledgments:** The authors would like to appreciate the anonymous reviewers for their constructive suggestions and comments to improve the quality of this paper.

**Conflicts of Interest:** The authors declare that they have no conflicts of interest.

**Data Availability:** The data used to support the findings of this study are included within the article.

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
