# Peer review of "Evaluation of the Hydraulic, Physical, and Mechanical Properties of Pervious Concrete Using Iron Tailings as Coarse Aggregates"

_applsci, doi:10.3390/app10082691_

Round 1

Author Response

请参阅附件。

Reviewer 2 Report

At the manuscript “Evaluation of the hydraulic and mechanical properties of pervious concrete using iron tailings as coarse aggregates” presented the effective porosity, measured porosity, dry density, compressive strength and permeability coefficient of pervious concrete were studied iron tailings as coarse aggregates.

The following issues need to be explained and clarified by the authors:

  • Citations in the manuscript are not very diversified, and about 50% of publications come from a same journal. The review of references should have been more diverse.
  • Why did the authors decide to use the "Constant falling-head permeability satup" and to test the flow rate with water flow to the sample from upwards? How is it assured that the same water inflow to the whole sample is provided? Please include a photograph of the used apparatus.
  • In Chapter 3 and its subchapters, the statistical methods should be applied and the reasons for their use should be indicated. In figures 4, 6, 7, 9 lack detailed designations and descriptions must be completed, for example the deviations shown in figure 9, which means and how to interpret them. In the manuscript, for example, figures 6 and 7 wrongly indicate the model fit measure as R2 should be R2. This should be checked everywhere in the article.
  • What are the boundaries of applicability and recommendations for the models indicated in the chapter 3 of the manuscript and quoted in the conclusions? These should be collated and the boundaries of applicability for each model defined.
  • The authors should include in the conclusions information on the possible practical application of the presented research results.
  • The elements of novelty in the article should be presented.

In view of the above, I regard this article as should reconsider after major revision.

Author Response

请参见附件。

Reviewer 3 Report

The article concerns an important topic connected to sustainable urban development—namely, the use of the iron tailings in the pervious concrete mixture. However, I have some comments and questions for the presented paper.

In my opinion, the introduction section is too short and doesn’t present every important aspect raised in the article. In the author’s research, they have differences between target, measured and effective porosity. They don’t provide any information about this issue in the introduction section. Did other scientists also have this problem? How they solved this issue? Moreover, iron tailings are recycled aggregate, so it is essential to present information about the impact of this aggregate on the environment. The authors described permeability properties. Thus, in my opinion, the introduction section should provide at least leaching properties of the iron tailings from the literature. This is the crucial information because if this aggregate is not environmentally safe, this is the reason why engineers don’t use it.

In the article, the authors used the parameter “porosity”; however, in Table 2 parameter name “void ratio” appeared. These parameters are different, so I think authors should also provide porosity of iron tailing in table 2.

The authors have done a lot of research. However, only the compressive strength tests were made according to the standard or norm. What about other made tests? The authors described their method of preparation processed for previous concrete. What is the standard procedure of preparation processed for previous concrete? The authors should point out the differences between their approach and standard procedure.

What were the Proctor hammer weight and fall height? What compaction energy was produced in the Proctor compaction process.

The authors presented differences between measured and effective porosity. In my opinion, these differences are caused by incorrect measurement methodology using a camera. Namely, the authors took a picture from the top, middle and bottom sections of the sample, but the authors photographed the horizontal plane of the sample. In that plane, porous size is not affected by the compacting process. In the vertical plane, the porous size will be much smaller due to anisotropy caused by Proctor compacting. Please comment on that issue.

I am also worried about the methodology of measuring the filtration coefficient. The authors created a water head difference 150 mm, while the sample height is 100 mm, so the hydraulic gradient is 1,5. This is a very high value and it does not reflect practical engineering cases. Please comment on that issue.

The blueprint of the constant falling-head apparatus is also striking. In such porous material like pervious concrete, measuring the permeability coefficient based on water flow from top to the button of the sample is incorrect because water easily finds a privileged filtration path and measurement will be not reliable. The permeability coefficient should be measured based on filtration from the button to the top of the sample, then we have certainty that the medium is fully saturated. Please comment on that issue.

In figures 4, 7, 8 and 9, the authors provided some statistical data presented on the graphs by whiskers. However, there is no explanation of what whiskers present. How many measurement authors used for the calculation of statistical parameters given by whiskers?   

Y-axis in figure 4 is not clear.

In line 201 the authors wrote: “It has been reported that the porosity has a decisive effect on the strength of concrete, at the same time, the pore structure (such as pore size, pore distribution, pore shape and orientation, etc.)”. How can you measure pore size, pore distribution, pore shape and orientation and what was pore structure of presented pervious concrete?

In my opinion, authors should propose an amendment for calculation of the target porosity, thanks to which obtained effective porosity will be closer to the assumed porosity.  

Round 2

Reviewer 2 Report

I would like to thank the authors for their answers. However, they do not fully answer the questions posed in the review. The issue of the semi-empirical formula (lines 298, 299, 301) needs to be clarified, whether there is any practical application for them, if so, it is necessary to indicate which ones and specify when and with what limitations they can be applied.

Reviewer 3 Report

Very interesting research. In your next tests please concern my previous comments and the article will be stunning. I wish you all the best. 

Author Response

Thank you for your encouragement, I will keep working hard.